# Performance and Mechanism of Asphalt Modified by Buton-Rock Asphalt and Different Types of Styrene-Butadiene-Rubber

**Fangting Qu [1], Songtao Lv [1,*], Junfeng Gao [2] and Chaochao Liu [1]**

[1] National Engineering Laboratory of Highway Maintenance Technology, Changsha University of Science & Technology, Changsha 410114, China; qft@stu.csust.edu.cn (F.Q.); lcccs@stu.csust.edu.cn (C.L.)
[2] South Erhuan Middle Section, College of Highway, Chang'an University, Xi'an 710064, China; gaojunfeng@chd.edu.cn
* Correspondence: lst@csust.edu.cn

**Abstract:** In this paper, two types of Styrene-Butadiene-Rubber (SBR) were adopted to modify the Buton-rock asphalt (BRA) modified asphalt, aiming to select a binder with excellent comprehensive performances. Powder SBR and latex SBR (0%, 2%, 4%, 6%, and 8%), were mixed with the 15 w% BRA modified asphalt. The characterization of rheological properties included dynamic shear rheometer, rotational viscometer, and bending beam rheometer test. The short-term aging performance was characterized by the ratio of the complex shear modulus from the un-aged and rolling thin film oven (RTFOT) -aged asphalt. Besides, Fourier transform infrared spectroscopy and scanning electron microscopy were conducted to reveal the modification mechanism. It was observed that the two kinds of BRA-SBR modified asphalt had preferable anti-crack capacity at low temperatures than the BRA modified asphalt. Compared with latex SBR, the powder SBR significantly improved the high-temperature performance, and the anti-aging capacity was stable. However, some negative influence occurred by the addition of latex SBR on the anti-rutting and short-term aging property. According to the micro-mechanism analysis, adding powder SBR and latex SBR into BRA modified asphalt was a physical blending process, and they improved the dispersion state of BRA in asphalt. Based on the comprehensive performance, the recommended combination was BRA and powder SBR.

**Keywords:** rheological properties; anti-aging capacity; performance contrast; compound modified asphalt; Styrene-Butadiene-Rubber (SBR); Buton-rock asphalt (BRA)

## 1. Introduction

Asphalt has been widely used as a binder. However, due to the increasing traffic loads and changeable environment, the pavement constructed with virgin asphalt always failed to achieve an expected service life [1,2]. Therefore, physical modification, chemical modification, or modification combining physical and chemical methods has been the favorable method to improve the road performance and durability of asphalt pavement [3–6].

A series of physicochemical modification studies have been successively carried out to verify the effect of the modifier on base asphalt and their mechanism. Luo et al. [7] found that Styrene-Butadiene-Styrene (SBS) could improve the high-temperature resistance to rutting, low-temperature crack resistance, fatigue resistance, and water stability of asphalt mixtures. Zhou et al. [8] compared the performance of Styrene-Butadiene-Rubber (SBR) modified asphalt with matrix asphalt and SBS modified asphalt. They found that SBR modified asphalt had the highest ductility. Zhong et al. [9] discovered that the bitumen binder and mixtures with the addition of Xinjiang rock asphalt were more competent in resisting permanent deformation, fatigue performance, and moisture

susceptibility. Li et al. [10] found that the aging rate of the rock asphalt modified mixtures was slightly lower by comparing the normalized ratio J'(t) (defined as the changing rate of creep compliance with time) of the mixtures before and after long-term aging. All the results showed that a single modifier improved the performance of asphalt to a certain degree. However, some other properties of asphalt could not be improved only by a single modifier, and the compound modified asphalt should be taken into consideration. As mentioned above, the SBS modified asphalt has lower anti-aging properties due to the presence of unsaturated carbon-carbon double bonds [7]. The anti-aging performance of SBR modified asphalt is lower due to structural instability [8]. The anti-cracking capacity at low temperatures was immolated because of the increased stiffness of rock asphalt modified asphalt [9]. The asphalt used as a binder must be qualified for a wide temperature range, moisture environment, and loading capacity because the service environment of asphalt pavement bears the effects of vehicle loading and environment factors, such as moisture, high and low temperature, simultaneously. Most of the single modified asphalt cannot balance the comprehensive road performance. Thus, it is highly important to finding a better combination of several materials to improve the performance of base asphalt.

In this regard, the compound modification method is often utilized to ameliorate the road performance of asphalt from multiple aspects. By adding Nano-clay and SBR, Ameri et al. [11] found that the rheological properties of base asphalt increased, and the temperature sensitivity decreased. In order to improve the compatibility, low-temperature, and high-temperature performance of SBR modified asphalt, Liang et al. [12] simultaneously added SBR and PPA to the matrix asphalt. Their results showed that PPA replenished the insufficiency of high-temperature performance and compatibility of SBR powder, and the good low-temperature performance provided by SBR powder was not affected by the addition of PPA. Cai et al. [13] combined the rock asphalt, Nano-silica, and SBS as modifiers, and they found that the compound modified asphalt had outstanding performances. Shi et al. [14] proposed the Nano-silica and rock asphalt modified asphalt. They found that the compound modified asphalt had better rheological performance and effective cost, and the good rheological property was partly attributed to presence of rock asphalt. It can be seen that the compound modification method is an effective way to obtain asphalt binder with good comprehensive performance.

From the above, the rock asphalt can fully exert its excellent high-temperature performance, and enhanced the anti-aging property after mixing with other modified asphalt. The SBR can also provide good low-temperature performance for the asphalt binders modified by other materials. Besides, compared with the widely used SBS modified asphalt, the superior processability, economic efficiency, and environmental benefits of rock asphalt and SBR can make them appropriate additives to enhance the performance of base asphalt [15–18]. Some researchers have verified the expedience of other natural asphalts and SBR as compound modified materials in asphalt, including gilsonite and Lack asphalt. Ren et al. [19] found that the addition of SBR powder to the gilsonite improved the low-temperature deformation capacity by decreasing the stiffness values of asphalt binder with opposed impact on the m-value (i.e., the change rate in stiffness to time). The results suggested that the temperature sensitivity and anti-deformation capacity at high temperatures improved after the compound modification. Liu et al. [20] investigated the compound modified effects of SBR and Trinidad Lack Asphalt on the base asphalt. The results demonstrated that compound modified asphalts have favorable low-temperature deformation property and high-temperature performance. Given that the above studies have achieved good results, the Buton-rock asphalt (BRA) and SBR were selected as modifying agents to mix with the matrix asphalt in the present study. The purpose is to combine the anti-crack capacity of SBR at low temperatures and the anti-deformation capacity at high temperatures and anti-aging capacity of BRA modified asphalt. Furthermore, two types of SBR, powder SBR and latex SBR, are commonly used as asphalt modifiers. Wang et al. [21] added latex SBR to 90 PG base asphalt to improve the low-temperature and anti-aging performance. Sharvin et al. [22] studied the difference between powder SBR and latex SBR on the performance of base asphalt. However, few studies have focused on the difference between powder SBR and latex SBR on the performance of

BRA modified asphalt, not just base asphalt. For obtaining the compound modified asphalt with better properties, two types of BRA modified asphalt containing powder SBR or latex SBR were compared in this study.

Therefore, powder SBR A and latex SBR B were mixed with BRA modified asphalt to prepare BRA-SBR modified asphalt in the high-speed shear equipment. The dosages of SBR were 0%, 2%, 4%, 6%, and 8%. The examination of the high-temperature performance of compound modified asphalt was based on the rotational viscometer (RV) and dynamic shear rheometer (DSR) tests. The bending beam rheometer (BBR) test was adopted to compare the difference of low-temperature performance of the compound modified asphalt, both BRA-SBR A and BRA-SBR B modified asphalt. The short-term aging capacity was evaluated by the rheological performance of un-aged and RTFOT-aged asphalts. Meanwhile, the modification mechanism of BRA modified asphalt before and after adding SBR was revealed by the Fourier transform infrared spectroscopy and scanning electron microscopy test.

## 2. Materials

### 2.1. Base Asphalt

The general properties of PG 70 base asphalt were tested, and the results are displayed in Table 1 which indicated that all technical performances met the requirements of ASTM standards.

**Table 1.** Technical properties of PG 70 base asphalt.

| Items | Test Results | Technical Requirements | Specification |
|---|---|---|---|
| Penetration (25 °C, 0.1 mm) | 68.2 | 60–80 | JTG E20 T0604 |
| Softening point (°C) | 49.1 | ≥46 | JTG E20 T0606 |
| Ductility (15 °C, cm) | >100 | >100 | JTG E20 T0605 |
| After RTFOT | | | |
| Mass loss (%) | 0.15 | ≤±0.8 | JTG E20 T0609 |
| Residual penetration ratio (25 °C, %) | 64 | ≥61 | JTG E20 T0604 |
| Residual ductility (15 °C, cm) | 8.1 | ≥6 | JTG E20 T0605 |

### 2.2. Buton-Rock Asphalt (BRA)

Buton-rock asphalt (BRA) was produced in Buton Island, Indonesia, and its bitumen content is 23.8%. The BRA was first crushed into fine particles and then passed through 0.15 mm square opening sieve. The conventional performance indicators of BRA met the requirements as shown in Table 2. In order to facilitate the dispersion of BRA particles in the asphalt, a drying procedure was performed to remove a small amount of water in BRA and loosen the blocky BRA.

**Table 2.** Technical properties of Buton-rock asphalt (BRA).

| Items | Test Results | Technical Requirements | Specification |
|---|---|---|---|
| Solubility of TCE (%) | 25.1 | >25 | JTG E20 T0607 |
| Water content (%) | 0.5 | <1.0 | JTG E42 T0332 |
| Ash content (%) | 74.9 | 65–75 | JTG E20 T0735 |
| Density (g/cm$^3$) | 1.75 | >1.6 | JTG E42 T0328 |

### 2.3. Styrene-Butadiene-Rubber (SBR)

Powder SBR and latex SBR, named SBR A and SBR B, respectively, were used in this study. They were provided by Mingji JinTai rubber and plastic products processing Co., Ltd., Tianjin, Hebei, China. Tables 3 and 4 present the properties of SBR A and SBR B, respectively.

**Table 3.** Properties of powder Styrene-Butadiene-Rubber (SBR).

| State | Butadiene Content (%) | Styrene Content (%) | Mooney Viscosity (1 + 4) 100 °C | Particle Size (Mesh) |
|---|---|---|---|---|
| Milky white | 70 | 23.5 | 52 | 40 |
| Volatile fraction (%) | Elongation at break | Ash (%) | Tensile strength (MPa) | Organic acid (%) |
| 0.2 | 540 | 7.5 | 26.5 | 6.3 |

**Table 4.** Properties of latex SBR.

| State | Viscosity (25 °C, mpa·s) | Solid Content (%) | Proportion | Average Molecular Weight (g/mol) |
|---|---|---|---|---|
| Pale yellow | 7000 | 60 | 0.92 | 50,000 |

### 2.4. Preparation of Modified Asphalt

The content of rock asphalt is 15% of the total mass of the base asphalt and SBR in this paper. The content of SBR was set as 0%, 2%, 4%, 6%, and 8% by weight of base asphalt. With the method of high-speed shearing, different types and contents of SBR and 15 w% BRA were added successively into the base asphalt to prepare modified asphalts. The production process of BRA-SBR compound modified asphalt is shown in Figure 1.

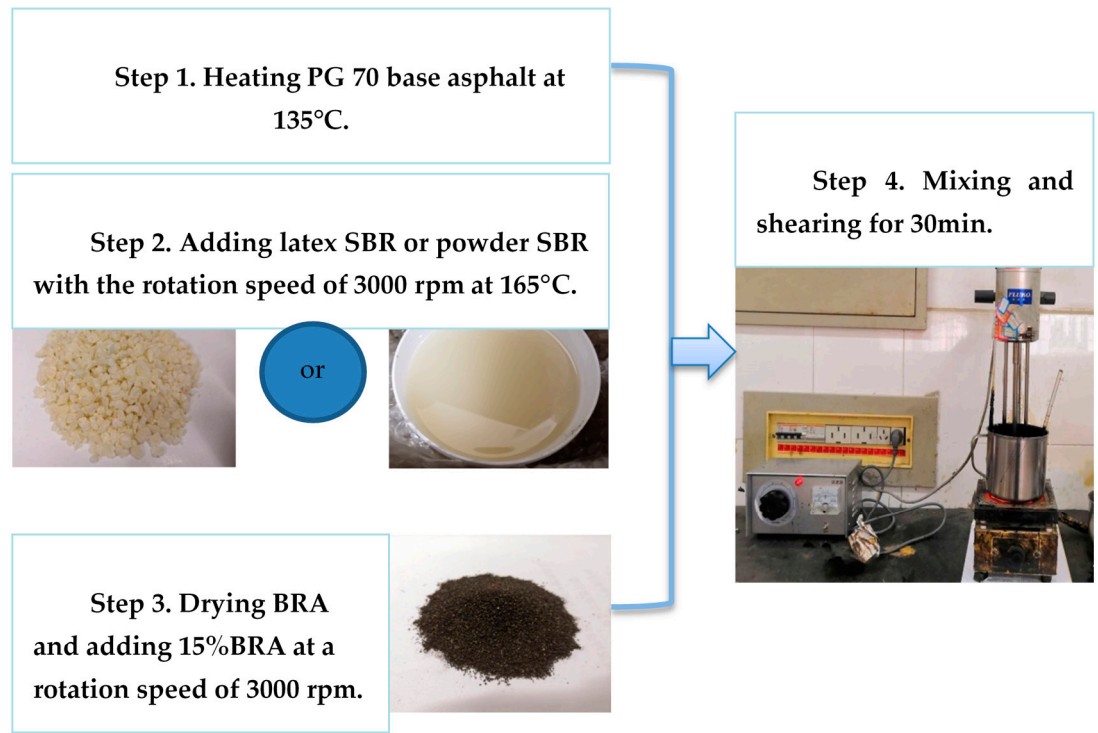

**Figure 1.** Production process of BRA-SBR compound modified asphalt 3. Test methods.

## 3. Tests

### 3.1. High-Temperature Performance Experiments

Referring to the AASHTO T316 and T315, the anti-deformation capacity of modified asphalt at high temperature is usually evaluated by rotational viscometer (RV) and dynamic shear rheometer (DSR) test. The RV test was conducted to determine the rotational viscosity by maintaining the rotational speed of 10 r/min at 135, 145, 165, and 175 °C, respectively. Each sample experienced three repetitive experiments. In addition, the phase angle and complex shear modulus of un-aged and aged specimens were also determined by the DSR test, which is usually used for measuring the capacity of resisting permanent deformation. The diameter of the oscillation plate is 25 mm, and the spacing

between the parallel plates was 1 mm. The vibration frequency was 10 rad/s. The initial temperature was 46 or 52 °C, and every 6 °C was an interval increased temperature.

### 3.2. Low-Temperature Performance Experiment

As is known from the low-temperature cracking mechanism, the resistance to temperature shrinkage cracking in asphalt pavement is influenced by the rheology characteristics of asphalt materials at low temperatures. SHRP proposed the bending beam rheometer (BBR) intending to evaluate the low-temperature performance of asphalt. Beam specimens were formed in a mold with a size of $127 \times 12.7 \times 6.35$ mm. Given that thermal cracking caused by rapid low-temperature change occurs during the service period of asphalt pavements, asphalt has experienced short-term and long-term aging at this time. The specimens were first subjected to long-term aging and then performed on the low-temperature test in the TE-BBR instrument at the temperature of −6 °C. Based on this, the creep curve slope (m-value) and stiffness modulus at 8, 15, 30, 60, 120, 240 s were obtained, and the result at the 60 s was chosen to measure the low-temperature properties of the BRA modified asphalt modified with SBR A and SBR B. The larger stiffness modulus of asphalt binder will cause increased susceptibility to cracking due to the brittleness. The smaller m-value, which is equivalent to the increase of the stiffness, could result in the deteriorated resistance to cracking at low temperatures. Thus, lower stiffness modulus is needed at a certain low-temperature condition, as well as a higher m-value.

### 3.3. Anti-Aging Property Experiment

The rolling thin film oven test (RTFOT) and pressure aging vessel test (PAV) can simulate the short-term and long-term aging behavior of asphalt binders. The specimens after short-term aging were prepared for the DSR test, and those after long-term aging were prepared for the BBR test in the present paper. All aged specimens for specific tests were treated according to the detailed procedure of ASTM D2872 and D6521-19a.

Based on the previous studies, the change in rheological properties of asphalt before and after aging could be utilized to evaluate anti-aging properties [23,24]. Therefore, the complex modulus aging index (CMAI) is used to characterize the anti-aging capacity in this study, which was calculated by the ratio of complex shear modulus before and after short-term aging [25]. Asphalt, which has a smaller CMAI, showed better aging performance.

### 3.4. Fourier Transform Infrared Spectroscopy (FTIR) Test

The various groups making up the material molecule have their own specific infrared characteristic absorption peaks. The fingerprint region (1300–500 cm$^{-1}$) and the functional region (4000–1300 cm$^{-1}$) in infrared spectra can be used to identify the functional groups. The modification mechanism is usually revealed by comparing the infrared spectra before and after the addition of the modifier in a visual way.

Fourier transform infrared spectroscopy (FTIR) includes traditional transmission FTIR mode and attenuated total reflection (ATR) FTIR mode [26]. In this paper, the KBr window method which is classified to traditional transmission mode was selected to perform the FTIR test. For every FTIR sample, the asphalt was 0.1 g, and the CCl$_4$ dissolving solution was 2 mL. The infrared spectroscopies of latex SBR, BRA modified asphalt, and two BRA-SBR modified asphalts were tested. The scan range was 4000~500 cm$^{-1}$.

### 3.5. Scanning Electron Microscope (SEM) Test

Scanning electron microscope (SEM) test is usually used to observe the micro morphology of asphalt. The effects of two types of SBR on the compatibility between BRA and asphalt were explored with the help of SEM test in this paper. The asphalt samples were sprayed with gold and then image scans were performed on BRA modified asphalt, BRA-SBR A modified asphalt, and BRA-SBR B modified asphalt with the magnification of 200.

## 4. Results and Discussion

### 4.1. High-Temperature Performance

Figure 2 shows the viscosity of BRA and BRA-SBR modified asphalt. As the temperature increases, the viscosity of all modified asphalt samples decreased, but the amplitude reduced. Increasing temperature from 135 to 145 °C or from 165 to 175 °C, the viscosity of asphalt modified with BRA and 2% SBR lowered 36.4% and 21.7%. As presented in Figure 2, at different experimental temperatures and SBR contents, the viscosity of asphalt modified with 15% BRA and SBR A was higher than that of BRA and BRA-SBR B modified asphalt. Viscosity is a measure of resistance to flow deformation at high temperatures. The highest viscosity value of BRA-SBR A asphalts meant the best flow deformation resistance among the three modified asphalts. It is well known that the introduction of rock asphalt could notably enhance the high-temperature performance of asphalt systems [6,27]. Additionally, when BRA was mixed with the SBR A, the high-temperature performance of asphalt would be further enhanced. This could be influenced by the enhanced stiffening properties of the asphalt binder due to the use of SBR A in BRA modified asphalt [19]. The viscosity increased more obviously as the SBR A content increased. Although higher viscosity may cause the increase of the mixing and compaction temperatures during construction, the viscosity measured at 135 °C could still be in line with the Superpave specification that the viscosity of asphalt binder should be no more than "3 Pa·s". However, for the BRA modified asphalt, the use of SBR B caused a slight decrease in the viscosity. The viscosity of BRA modified asphalt was slightly affected by the amount of SBR B. The high-temperature fluidity of BRA modified asphalt was influenced slightly by SBR B.

The high-temperature performance of asphalt binders modified with two materials was further compared by the rheological parameters of the DSR test. The phase angle of un-aged specimens is presented in Figure 3, which is usually used for reflecting the viscoelastic characteristics of materials. The smaller the phase angle, the more elastic components and the less viscous components exist in the asphalt. The increase in the elastic portion is more conducive to the enhancement of the deformation resistance capacity [28]. As Figure 3 shows, the mixed addition of 15% BRA and SBR A lowered the phase angel of BRA modified asphalt. The smaller phase angle of compound modified asphalt manifested that the addition of SBR A made it possible for asphalt binder to achieve more elastic recovery when resisting shearing. The increase in the elastic network and reduction in the viscous network increased the high-temperature performance of the asphalt binder. On the contrary, the phase angle curves of SBR B-BRA modified asphalt were located higher than that of BRA modified asphalt. SBR B decreased the elastic property of BRA modified asphalt.

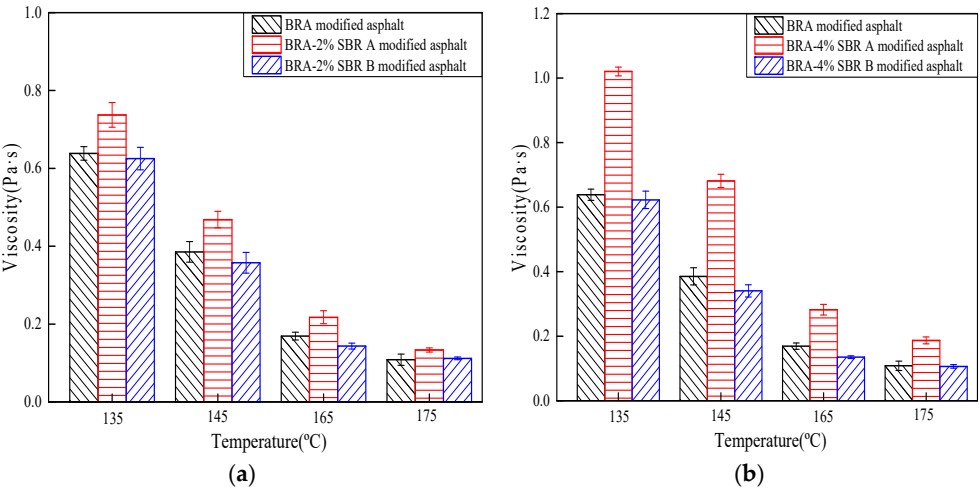

**Figure 2.** *Cont.*

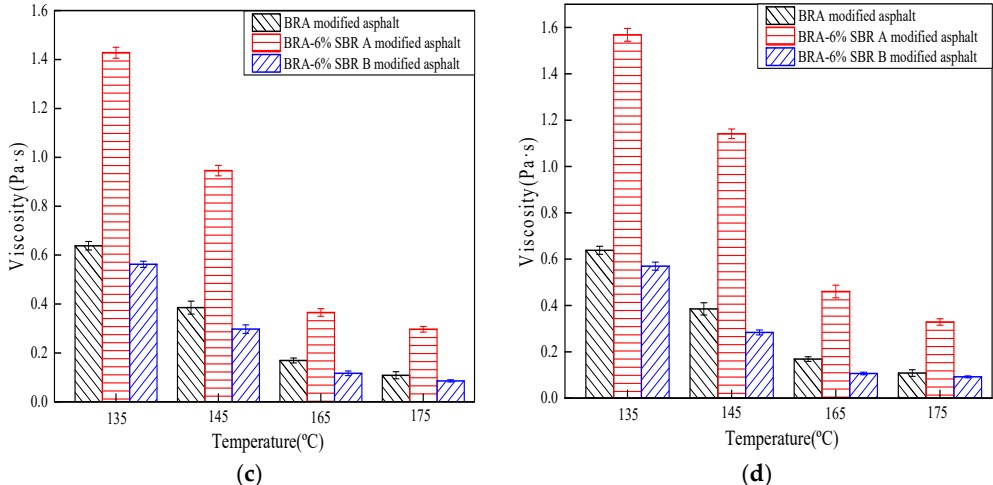

**Figure 2.** Viscosity of specimens. (**a**) BRA and 2%BRA-SBR modified asphalt; (**b**) BRA and 4%BRA-SBR modified asphalt; (**b**) BRA and 6%BRA-SBR modified asphalt; (**d**) BRA and 8%BRA-SBR modified asphalt.

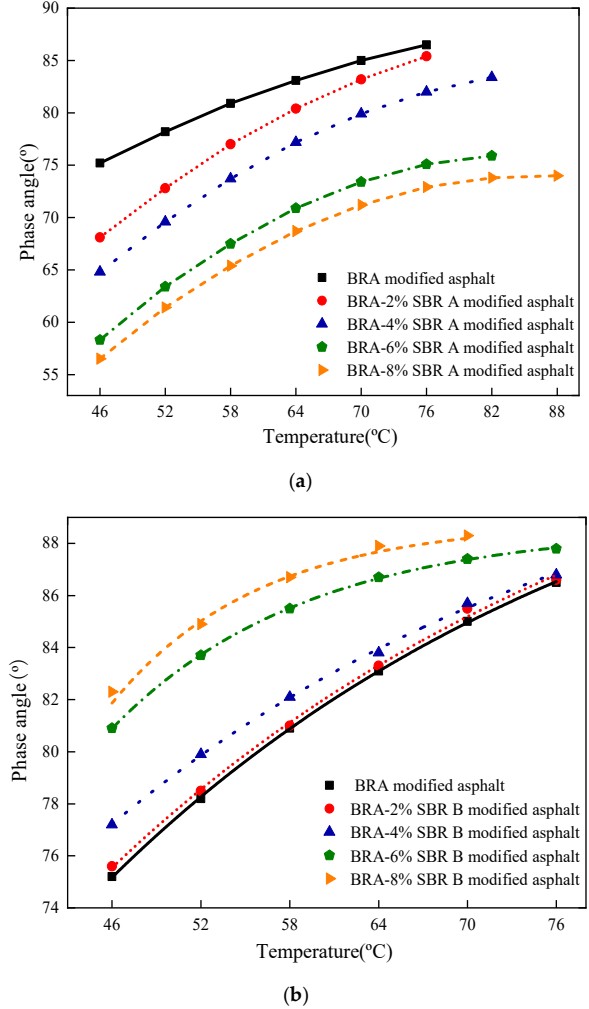

**Figure 3.** Phase angle of specimens. (**a**) BRA and BRA-SBR A modified asphalt; (**b**) BRA and BRA-SBR B modified asphalt.

Figure 4 shows the variation of the complex shear modulus of asphalt. Both SBR A-BRA and SBR B-BRA compound modified asphalt displayed a rapid decline in complex modulus as the experimental temperature increased, implying the deterioration of high-temperature deformation resistance in asphalt. This may attribute to the reason that asphalt has a transformation from a high-elastic state in low temperature to a high-viscous state in relatively higher temperatures [29]. The raise in temperature caused the volume of asphalt to expand. The spacing of molecules increased, and the random movement of molecules intensified. The displacement is more prone to beget under the action of external forces, reducing the ability of asphalt to resist external forces. Apart from the temperature, the type of modifier also has an effect on high-temperature performance. As Figure 4 shows, the complex shear modulus of BRA-SBR A modified asphalt was higher than that of BRA modified asphalt.

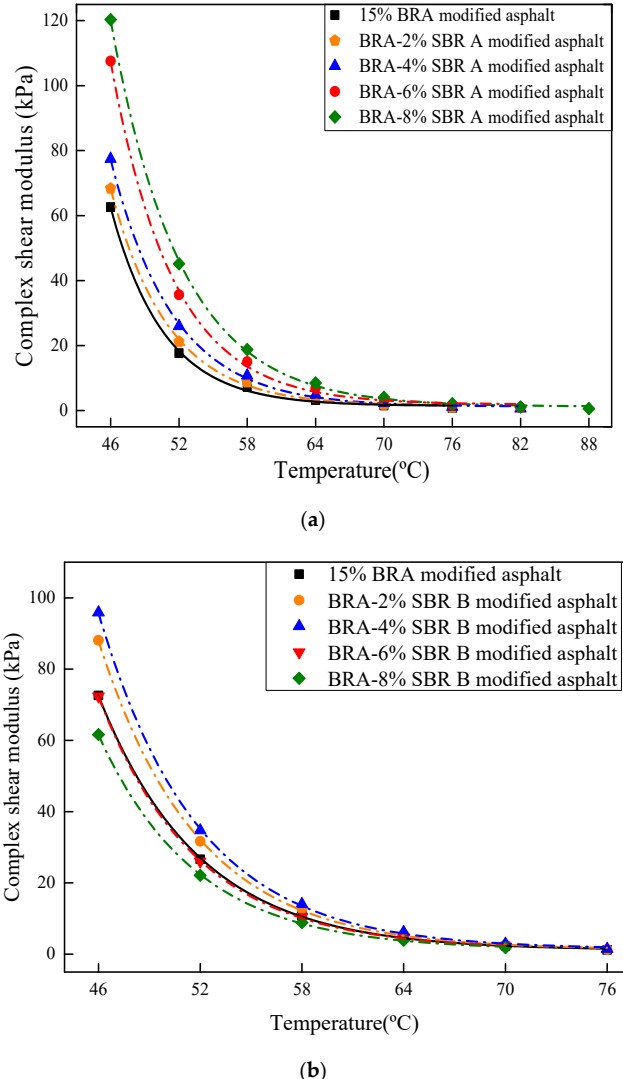

**Figure 4.** Complex shear modulus of specimens. (**a**) BRA and BRA-SBR A modified asphalt; (**b**) BRA and BRA-SBR B modified asphalt.

Moreover, the complex shear modulus of BRA modified asphalt was significantly enhanced as the SBR A content increased. For example, increasing the content of SBR A to 8% at 64 °C, the complex shear modulus of BRA-SBR A modified asphalt increased 2–3 times that of asphalt binder modified with BRA. A similar conclusion could be obtained that the anti-deformation capacity of BRA modified asphalt at high temperatures was greatly improved by the introduction of SBR A. For BRA-SBR B compound modified asphalt, with the increase in SBR B, the shear modulus increased firstly and then decreased.

It manifested that the anti-deformation capacity of SBR B-BRA modified asphalt could be comparable to that of the BRA modified asphalt just within a certain range of SBR B content. The high-temperature resistance to rutting was still less satisfactory when the SBR B content reached 6%.

### 4.2. Low-Temperature Performance

Figure 5 shows the creep curve slope (m-value) and creep stiffness of BRA and BRA-SBR modified asphalt at −6 °C. Previous research showed that the addition of BRA could significantly weaken the low-temperature performance of asphalt binders. The asphaltene concentration of asphalt binder increased due to the high asphaltene content of BRA [19]. Hardening effect could adversely affect the low-temperature properties of asphalt. The experimental results of Figure 5 showed that the creep stiffness of the two compound modified asphalts were lower compared with the BRA modified asphalt, and the m-value upper. This meant that when the temperature dropped to a lower value, adding two types of SBR into BRA modified asphalt made it difficult to accumulate tensile stress due to temperature shrinkage. The possibility of low-temperature cracking was then reduced. Unlike the BRA, two types of SBR could soften the asphalt binder, and improve the low-temperature performance. The network structure of SBR could overcome drawbacks of the low-temperature performance of asphalt binders with BRA.

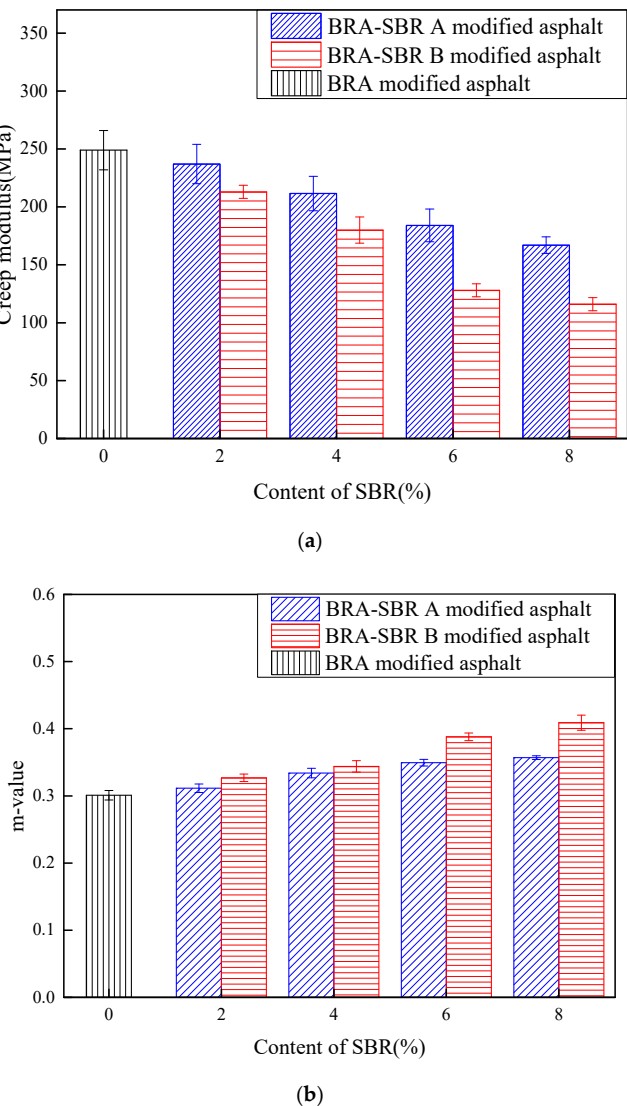

**Figure 5.** Creep stiffness and m-value of specimens at −6 °C. (**a**) Creep modulus; (**b**) m-value.

On the other hand, as presented in Figure 5, the two kinds of compound modified asphalts presented the more pronounced decline in creep stiffness and an increase in m-value in the wake of the increase of SBR content. That is to say that the higher the addition contents of SBR, the more noticeable the improvement effect on anti-crack capacity at low temperatures. However, there are still some differences in the improvement extent in low-temperature performance. The compound modification effect on low-temperature cracking was superior because of the existence of BRA and SBR B at the same time. When the content of SBR B was 2%, 4%, 6%, and 8%, the low-temperature flexibility of compound modified asphalt was 1.09, 1.11, 1.29, and 1.36 times of 15% BRA modified asphalt at −6 °C, respectively. Under the same low-temperature condition, the addition of SBR B resulted in a more significant improvement in stress relaxation and flexibility for the compound asphalt than the introduction of SBR A.

### 4.3. Short-Term Aging Capacity

The aging behavior tends to change the asphalt materials' physical and chemical properties, which are conversely reflected in its rheological properties [30–34]. Figure 6 illustrates the CMAI of base asphalt and asphalt modified with BRA and SBR A. It was conspicuous that when the temperature was less than 64 °C, the BRA-SBR A modified asphalt tended to result in a lower CMAI value than base asphalt. It indicated that the rheological properties of BRA-SBR A modified asphalt were not as easily affected by short-term aging as base asphalt. The anti-aging performance of base asphalt was improved by the modification of BRA and SBR A. Above the temperature of 64 °C, the compound modified asphalt has a comparable CMAI value to that of the base asphalt.

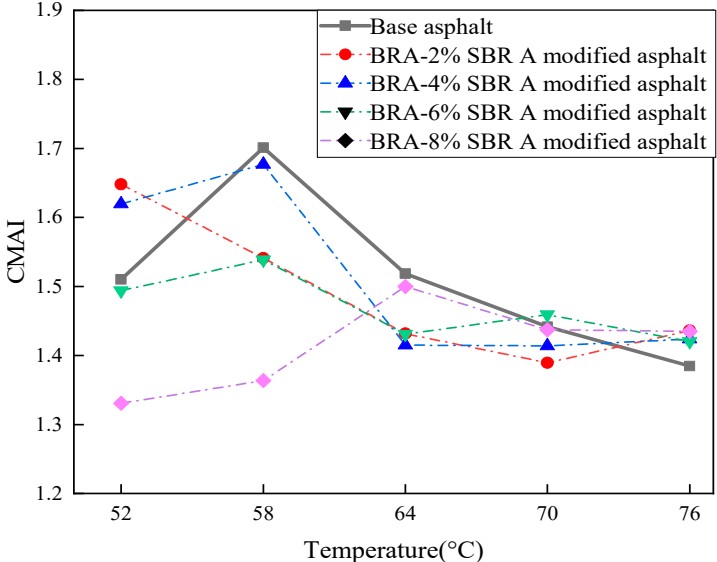

**Figure 6.** The complex modulus aging index (CMAI) of base asphalt and BRA-SBR A modified asphalt.

Figure 7 illustrates the CMAI of base asphalt and BRA-SBR B modified asphalt. As presented in Figure 7, the CMAI value of the base asphalt was merely affected by the BRA and SBR B modifiers when the SBR content was less than or equal to 6%. When the dosage of SBR B reached 8%, the CMAI value of the modified asphalt exceeded that of base asphalt. The introduction of rock asphalt could enhance the aging resistance of base asphalt, according to previous research [35]. However, at this time, the anti-aging performance of BRA-SBR B modified asphalt was even slightly worse than that of base asphalt. This showed that the addition of SBR B to BRA modified asphalt has a negative effect on short-term aging performance. The increase in anti-aging capacity caused by rock asphalt is not enough to offset the decrease caused by SBR B.

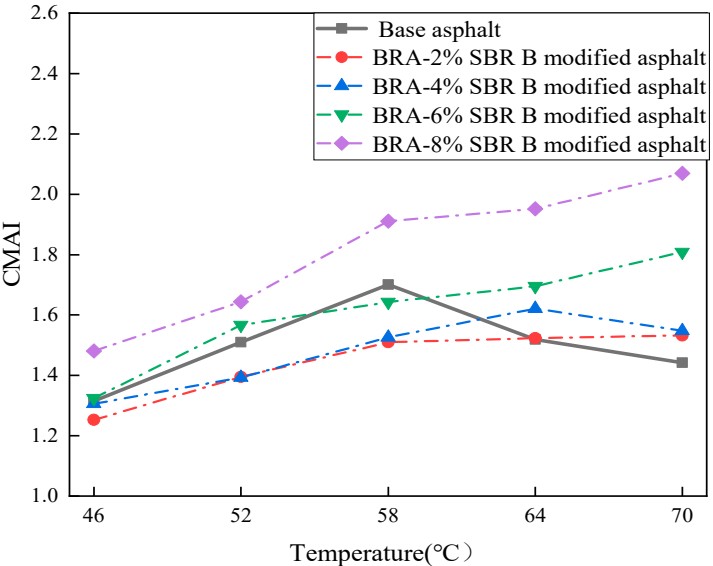

**Figure 7.** CMAI of base asphalt and BRA-SBR B modified asphalt.

### 4.4. FTIR Results

Figure 8 shows the FTIR spectra of BRA modified asphalt before and after modifying by 4% SBR A. The characteristic functional groups of BRA modified asphalt are listed in Table 5. It was clear from Figure 9 that the BRA-4% SBR A modified asphalt showed almost the same peak position with the BRA modified asphalt, apart from the 968 cm$^{-1}$ peak. With the incorporation of SBR A into the BRA modified asphalt, this characteristic peak derived from Trans-C–H wagging vibration in the butadiene block was brought in [36].

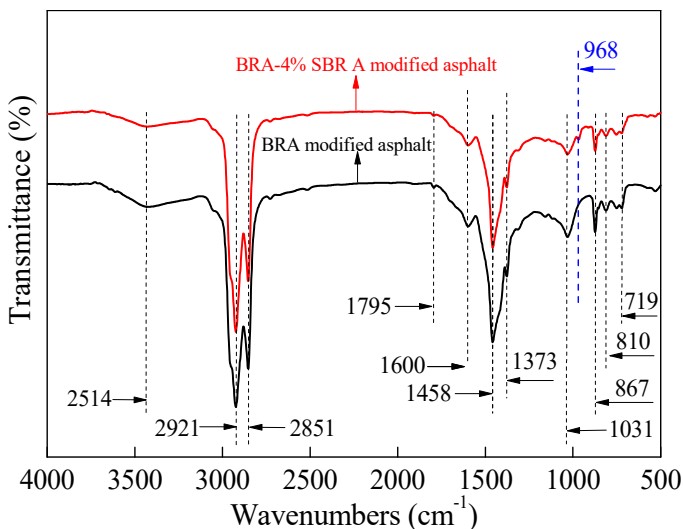

**Figure 8.** Fourier transform infrared spectroscopy (FTIR) of BRA modified asphalt before and after modifying by 4%SBR A.

Figure 9 shows the FTIR spectra of BRA modified asphalt before and after modifying by 6% SBR B. Compared with the BRA modified asphalt, the difference of BRA-SBR B modified asphalt in absorption peak position was located in 1730 cm$^{-1}$. As presented in Figure 9, the functional groups of 1730 cm$^{-1}$ derived from the strong absorption peak of SBR B, which represents the C = O stretching vibration methyl stearate.

**Table 5.** Functional group of BRA modified asphalt.

| Wavenumber | Functional Group |
|---|---|
| 2921 cm$^{-1}$ | C-H anti-symmetric stretching vibration in CH$_2$ |
| 2851 cm$^{-1}$ | C-H anti-symmetric stretching vibration in CH$_2$ |
| 1795 cm$^{-1}$ | C = O stretching vibration |
| 1600 cm$^{-1}$ | C = C stretching vibration |
| 1458 cm$^{-1}$ | C-H in-plane bending vibration |
| 1373 cm$^{-1}$ | C-H in-plane bending vibration in CH$_3$ |
| 1031 cm$^{-1}$ | S = O stretching vibration |
| 867 cm$^{-1}$/810 cm$^{-1}$/719 cm$^{-1}$ | C-H out-of-plane bending vibration |

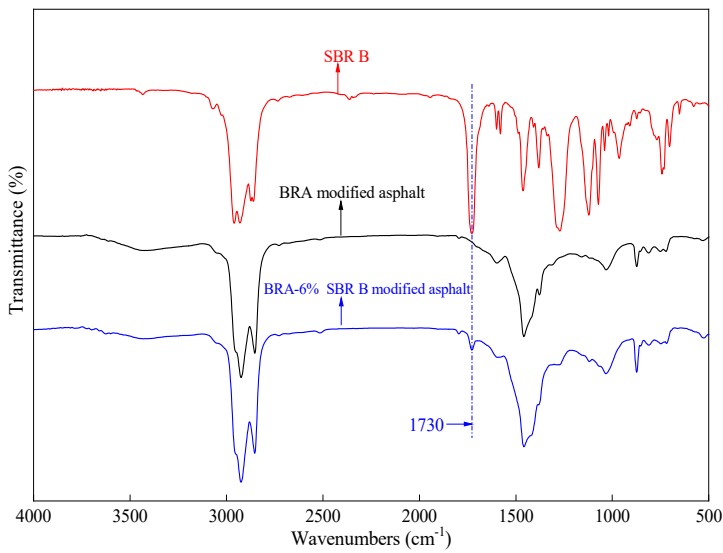

**Figure 9.** FTIR of BRA modified asphalt before and after modifying by 6%SBR B.

Therefore, the different functional groups of two BRA-SBR modified asphalt and BRA modified asphalt were derived from SBR additives. There was no new functional group generated. Whether it was SBR A or SBR B, adding them into BRA modified asphalt is a physical bending process.

In addition, the sulfoxide index of BRA modified asphalt, BRA-SBR A modified asphalt and BRA-SBR B modified asphalt was calculated. The ratio of peak area of wavenumber around 1031 cm$^{-1}$ and reference peak area (the sum of the peak areas around 1373 and 1458 cm$^{-1}$) was used to obtain the sulfoxide index [37]. The smaller the sulfoxide index, the weaker the aging affect. The sulfoxide index of BRA modified asphalt, BRA-SBR A modified asphalt, and BRA-SBR B modified asphalt were 0.061, 0.047, and 0.070, respectively. Therefore, the addition of SBR A could reduce the aging effect of BRA modified asphalt caused by the high temperature preparation process, while the SBR B could not.

## 4.5. SEM Results

During the preparation of BRA modified asphalt, rock asphalt particles swell with volume expansion and eventually dissolve in the asphalt. The ideal state is that all the rock asphalt particles are dissolved in the asphalt to form a stable homogeneous blending system with the asphalt. However, it can be seen from Figure 10a that there are incompletely dissolved BRA particles in the BRA modified asphalt. The reason may be that the dissolution of rock asphalt and base asphalt requires a certain time and a higher temperature [38]. Insufficient solubility of rock asphalt results in insufficient improvement of its modification effect, and the presence of conglobate rock asphalt particles easily lead to stress concentration, which seriously affects the mechanical properties of modified asphalt. From Figure 10b,c, it can be seen that with the addition of SBR, the agglomeration phenomenon in BRA modified asphalt is significantly reduced, and the compatibility of rock asphalt and base asphalt was significantly

improved, which is consistent with Ren's research [19]. In addition, the improvement effect of SBR B on the compatibility of rock asphalt was better than SBR A, which can explain that SBR B-BRA has better low temperature performance than SBR A-BRA modified asphalt.

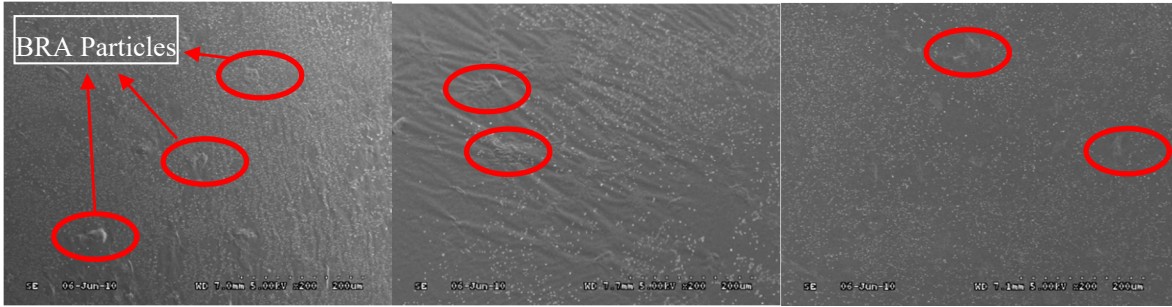

| (**a**) BRA modified asphalt | (**b**) BRA-SBR A modified asphalt | (**c**) BRA-SBR B modified asphalt |

**Figure 10.** Scanning electron microscope (SEM) images of BRA modified asphalt before and after modifying by SBR B.

## 5. Conclusions

In this study, two types of SBR, powder SBR and latex SBR, were utilized to achieve the improved properties of asphalt modified with BRA, respectively. The anti-crack capacity at low temperatures, anti-deformation capacity at high temperatures, and anti-aging capacity of compound modified asphalt were compared and evaluated. The modification mechanism of two types of SBR on the BRA modified asphalt was also revealed. Some conclusions are drawn from test results.

1. Both the powder SBR and latex SBR could significantly enhance the anti-crack capacity of BRA modified asphalt at low temperatures. With the increase in SBR content, the positive effect was more obvious. However, compared to the powder SBR, the addition of latex SBR could increase the low temperatures cracking performance to a greater extent.
2. The good high-temperature deformation resistance and anti-aging capacity of BRA modified asphalt were increased by the addition of powder SBR. The enhanced effect was more obvious with the increasing powder SBR content. However, the high-temperature anti-deformation capacity and anti-aging capacity of BRA modified asphalt declined slightly due to the addition of latex SBR.
3. After the addition of two types of SBR, there was no new functional group generated in the infrared spectra of BRA modified asphalt. The BRA, powder SBR or latex SBR, and base asphalt composed a physical blending system.
4. SBR could attribute the compatibility between BRA and base asphalt, and the facilitation of latex SBR was better than powder SBR.

Based on the above test results, BRA and powder SBR can be considered a suitable combination for improving the characteristics of base asphalt. The BRA-powder SBR compound modified asphalt could possess better high-temperature, low-temperature, and anti-aging performance than that of BRA modified asphalt. Additionally, the modification was a physical process.

**Author Contributions:** S.L. and F.Q. came up with the frame and designed the experiments; F.Q. and C.L. carried out the experiment; F.Q. completed the original draft preparation; S.L., J.G. and C.L. reviewed and edited the paper. All authors have read and agreed to the published version of the manuscript.

**Funding:** This work was funded by National Natural Science Foundation of China (No.51578081, 51608058), The Ministry of Transport Construction Projects of Science and Technology (2015318825120), The Projects of Transportation Science and Technology of Hunan (201701), Key Projects of Hunan Province-Technological Innovation Project in Industry (2016GK2096).

**Conflicts of Interest:** The authors declare no conflict of interest.

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
