# Peer review of "Performance and Mechanism of Asphalt Modified by Buton-Rock Asphalt and Different Types of Styrene-Butadiene-Rubber"

_applsci, doi:10.3390/app10093077_

Round 1
Reviewer 1 Report
In table 1 the properties of base asphalt are represented in terms of Chinese standards, that are little known, it would be better to explain how these standards differ from the more known ASTM or CEN standards.
Authors should give some more details on the BRA, because of different situations during its sedimentation process, the content of bitumen in the rock varies.
Page 5 is blank, it's probably a mistake.
Line 169 - The authors should explain how the ageing procedures used (T0610-2011 and T0630-2011) differ from the original ones (ASTM D2872 and D6521-19a).
In Figure 2 the additive SBR-B is mentioned twice, it is suggested to the authors to check, as there is probably an error (the red bar is the modification with SBR-A!).
Reviewer 2 Report
The article entitled "Performance and Mechanism of Asphalt Modified by Buton-Rock Asphalt and Different Types of Styrene-3 Butadiene-Rubber" is the next stage in the development of modification of asphalt mixes through appropriate effective modifiers. The article contains a lot of interesting information. Nevertheless, in the article there are several issues that need to be considered:
- How did the authors determine the optimal settings for the modification process such as: bitumen temperature, mixing time and additionally shear speed? It is known that a change in any of them results in different levels of components homogenization and different rheological properties of final bitumen. At the beginning of production process every modified bitumen requires this stage of production previously;
- In line 184 the authors used the KBr method. Can the authors provide references to literature where comparable results were obtained with another method, e.g. ART, where the chemical changes of bitumen’s were also studied;
- In line 223-224 should be rearranged. Which means "... to perform well ..". "well" of what? This is too general approach;
- Line 226. The increase in elastic recovery also applies to other trials of loading (not only shearing), eg "tensile test".
- The descriptions in the legends in Fig. 2 repeat twice. For example, in Figure 2a is "BRA modified asphalt" and twice given "BRA-2% SBR B modified asphalt";
- In fig. 4 should be "KPa" or "kPa"?;
- In Figure 4b and Figure 5. Did the results relate to a single sample measurement or mean value? If the mean, the range of results dispersion should be given (as in Fig. 2);
- The authors did not perform a quantitative analysis of the peaks of spectroscopic spectra. Nevertheless, a significant increase in the peak representing a wave number of about 1030 cm-1 (S = O) was observed in Figure 9. Is this not an indication of the aging effect caused by the high temperature mixing process ?
Round 2
Reviewer 1 Report
I honestly do not understand how a test procedure can be formulated on the basis of national conditions ('are more in line with national conditions of China'). The test standards normally reflect technical knowledge and may differ either in terms of parameters or how a parameter is measured. The test standards used by the authors are little known and although similar to the corresponding ASTM standards (according to the authors) it is not clear what the differences are. This makes research results of little use to the international scientific community.
Author Response
Please see the attachment.

This manuscript is a resubmission of an earlier submission. The following is a list of the peer review reports and author responses from that submission.
Round 1
Reviewer 1 Report
The novelty and contribution to knowledge still require analytical arguments with microscopic images. Empirical testing and validation is not sufficient for claiming high standards of contribution as set by Journal Applied sciences.
Reviewer 2 Report
The text should be improved by someone proficient in English
The presentation and descriptions of materials and methods are not in a scientific manner and should be improved significantly. There is a lot of speculation on the results that is not supported by the data. I recommend sticking to the data which are nice and discussing what you have.
Intro: define rock asphalt and J’(t)
What is multi field coupling?
Found a better combination?
What do you mean by processability?
Define m value
Modifying agent
What is 90#?
What does powder and latex SBR mean? How are they different from each other?
What is BRA?
Section 2.4 did you heat the base asphalt?
I am not sure I understand your preparation method: you heated the bitumen (T?) added BRA then 15 percent of this mass in SBR?
Is the latex a liquid? What is the particle size of the powder?
3.1: interval for what?
Fig 2 it would be helpful to show the base binder also DSR Sample size, frequency, temperature missing
3.3 what is short term aging and long term aging?
Why are short term aged samples used for DSR and long term BBR?
KBr?
Fig 2: What ist he scatter in the figures, error bars? Without this it is difficult to judge if the viscosity is indeed higher at higher temperatures. You cannot say that there is a deformation resistance with just the viscosity results, The most you can say is that there is a relationship there.
Discussion fig 3 you mention elastic portion what is that referring to? You need to define what you mean by performance. The phase angle data is saying something about the material behavior that is different, what is that exactly? Please see the relevant literature. What does perform well in phase angle mean?
Where do you show this: The experimental result of Figure 5 showed that, both the creep stiffness of the two compound modified asphalt were lower compared with the BRA modified asphalt, and the m-value upper.
Fig 5 is showing BRA-SBR data only
Paragraph 4.2 is not based on the results shown.
Fig 6: at what frequency was CMAI calculated?
Fig 8 and fig 9 it would be helpful to show FTIR of aged binders
Where did you show the Long term aged results?
Round 2
Reviewer 1 Report
Comments are not addressed but agreed to pursue in future work.
Reviewer 2 Report
cant see this: Three repetitions have been performed in the rotation viscosity test, and the error bars of the results have been added in Figure 2